# The Impact of an Online Mindfulness-Based Practice Program on the Mental Health of Brazilian Nurses during the COVID-19 Pandemic

**DOI:** 10.3390/ijerph20043666

**Published:** 2023-02-18

**Authors:** Edilaine Cristina da Silva Gherardi-Donato, Kranya Victoria Díaz-Serrano, Marina Rodrigues Barbosa, Maria Neyrian de Fátima Fernandes, Walusa Assad Gonçalves-Ferri, Elton Brás Camargo Júnior, Emilene Reisdorfer

**Affiliations:** 1Ribeirão Preto College of Nursing, University of São Paulo, Ribeirão Preto 14040-902, Brazil; 2School of Dentistry of Ribeirão Preto, University of São Paulo, Ribeirão Preto 14040-904, Brazil; 3Faculty of Medicine, Federal University of Uberlândia, Uberlândia 38400-902, Brazil; 4Nursing Department, Federal University of Maranhão, Imperatriz 65915-240, Brazil; 5Ribeirao Preto Medical School, University of Sao Paulo, Ribeirão Preto 14049-900, Brazil; 6Faculty of Nursing, Rio Verde University, Rio Verde 75901-970, Brazil; 7Department of Professional Nursing and Allied Health, Faculty of Nursing, MacEwan University, 10700 104 Ave NW, Edmonton, AB T5J 4S2, Canada

**Keywords:** mental health, nursing, health professionals, mindfulness, COVID-19, stress, depression, anxiety

## Abstract

This quantitative, before-after study was developed to evaluate the usefulness of an online mindfulness practices program to help nursing professionals deal with stress in the challenging context of the COVID-19 pandemic through the assessment of perceived stress, anxiety and depression, levels of mindfulness, and participants’ satisfaction with the program. Eligible participants were assessed at baseline to receive the online mindfulness training program for eight weeks and were appraised again at the end of the program. Standardized measures of perceived stress, depression, anxiety, and one-dimensional and multidimensional mindfulness were performed. Participant satisfaction was also studied. Adherence to treatment was 70.12%. The perceived stress, depression, and anxiety scores were significantly lower after the intervention. The mindfulness measure increased significantly, as well as the sense of well-being and satisfaction with life, study, and/or work. The participants showed high satisfaction with the program and would recommend it to other professionals. Our results indicate that mindfulness-based interventions represent an effective strategy for nurses in the face of the need for self-care with mental health and mechanisms that guarantee the sustainability of their capacities to continue exercising health care.

## 1. Introduction

The healthcare scenario during the ongoing COVID-19 pandemic represents a remarkable time when there is an urgent need to ensure that physical and mental demands do not cause lasting damage to healthcare professionals [1]. In a systematic review, researchers recognized healthcare professionals (HP), especially nurses, as a fragile population whose physical and mental health was overwhelmed during the COVID-19 pandemic. Psychological conditions, such as fear, psychological suffering, anxiety, depression, exhaustion, and lack of sleep, are some of the problems that can affect these professionals. According to this review, interventions aimed at stress management and innovative approaches to promoting psychological well-being should become a priority to minimize the impact of the pandemic on the mental health of health professionals, which is a vulnerable group needing support [2].

Recommendations on measures to protect the mental health of teams include a forthright acknowledgment of adversity, provision of external support and well-being monitoring, provision of well-being support, and evidence-based therapeutic interventions as and when necessary [1]. Among the possible interventions postulated in view of the main demands of health professionals, mindfulness-based practices have been appointed [3,4], which may be applicable to a wide variety of health conditions, as well as in diverse contexts which may be applicable to a wide variety of health conditions, as well as in diverse contexts [5]. They are indicated as therapeutic interventions with scientific evidence for reducing burnout, regulating emotions, improving emotional resilience, and reducing depression among nurses and other HP [5,6,7]. For nurses, the practice has great potential for improving quality of life and stress management [8].

In this sense, a state of mind that allows for more conscious choices and that reduces the perpetuation of dysfunctional behavior patterns becomes essential. Thus, mindfulness—understood as a skill that allows the expansion of the space between a stimulus and the response [9]—is essential in making decisions that precede the action. It is one of the qualities to be trained and improved in order to promote the maintenance of the balance of healthcare professionals, optimizing clinical practice and healthcare. Reported data established correlations between mindfulness practice and the care and satisfaction of patients assisted by health professionals [10]. A cross-sectional study carried out during the pandemic showed that the level of dispositional mindfulness of health professionals was negatively correlated with symptoms of anxiety and depression [11], indicating that the implementation of practices that improve the level of mindfulness can favor the maintenance of mental health in these professionals.

Studies concerning the effects of mindfulness-based interventions for HP have shown sustainable benefits for this population. Emergency health professionals undergoing a mindfulness intervention adapted from the traditional Mindfulness-Based Stress Reduction (MBSR) program, called “Mindfulness for Healthcare Providers” showed increased levels of satisfaction with compassion; increased in scores related to the mindfulness trait; a decrease in burnout levels, and a decrease in perceived stress [12]. However, the participation of nurses in a face-to-face mindfulness program is a challenge due to a lack of time. During the pandemic, this challenge became more pronounced and added to socializing restrictions.

Thus, during the COVID-19 pandemic, the eight-week Mindfulness Program, already offered in person by a service linked to a public university as an extension and research activity [13,14], was organized in an online format for nurses who were working in assistance in the pandemic context. The study of the intervention carried out aimed to establish a starting point to assess the feasibility and usefulness of the program as a support strategy to minimize the negative impact of challenging situations, such as the COVID-19 pandemic, on the mental health of nursing professionals.

The objective of this study was to assess the usefulness of an online mindfulness practices program to help nursing professionals with stress in the challenging context of the COVID-19 pandemic through the assessment of perceived stress, anxious and depressive symptomatology, levels of mindfulness, and participants’ satisfaction with the program.

## 2. Materials and Methods

### 2.1. Study Design

This study was an open-label, uncontrolled, before-after study, with measurements at baseline and immediately after the intervention. Data were obtained by completing electronic forms before and after the intervention. All participants in the sample sent their data within a period of seven days before the start of the eight-week intervention and again within the same seven-day interval after the intervention ended. The initial assessment took place in April 2020, and the final post-intervention assessment took place in June 2020.

### 2.2. Recruitment and Sample

This is a non-probabilistic convenience sample, which included: nursing professionals (nurses, technicians, and assistants) who assisted in any health services in Brazil during the COVID-19 pandemic and expressed interest in participating in the online training program for stress reduction and mental health promotion, based on mindfulness. Professionals who reported being in the acute phase of untreated psychopathological conditions were excluded from the sample, according to the guidelines of the intervention protocol. The recruitment of participants was made possible through the dissemination of the research on social media (Facebook and Instagram) and institutional emails (health services from the public assistance network).

An invitation was made to nursing professionals who were working in health care to participate in the online program entitled “Practicing Mindfulness during the COVID-19 pandemic: Developing skills to deal with stress and maintain mental health in challenging times”. Interested professionals filled out a pre-registration form with relevant data to verify eligibility criteria. Eligible professionals were invited to participate in an online session for guidance and explanation about the research and ethical aspects with the researchers, and they could express their agreement and availability to participate in the study. The recruitment of research volunteers took place from March to April 2020.

Among the 122 professionals who expressed interest in participating in the study, 45 were excluded; among them, 16 were ineligible (they were in the acute phase of psychopathological conditions and without treatment), four answered the instruments in duplicate, and 25 did not complete the assessment instruments of the study, completing the sample with 77 professionals who were included in the program for mindfulness practice. At the end of the eight-week intervention, 54 professionals completed the program, and 44 professionals performed the post-test, which constituted the sample analyzed in the present study (Figure 1).

### 2.3. Intervention

The intervention consisted of an eight-week conventional training program in mindfulness practice, based essentially on cognitive and emotional training to maintain more positive, conscious, and compassionate mental states, capable of causing psychosocial and physiological changes in concordance with better results concerning mental health, quality of life and psychological well-being [15,16,17,18]. It was conducted by a nurse who had worked in the field of mental health for 20 years and who had experience in applying a training program for mindfulness practice, with the collaboration of four health professionals (a psychologist, a dentist, a nutritionist, and a physical therapist), who are senior mindfulness instructors certified to teach the practices and have the experience to lead the program. In this way, the instructors responsible for the intervention tested in the present study provided training for mindfulness practices in a secular context, maintaining alignment with the foundations of the Mindfulness-Based Stress Reduction program (MBSR) of Kabat-Zinn [19] and the recommendations from the UK Network for Mindfulness-Based Teachers Good Practice Guidelines for Teaching Mindfulness-based Courses [20].

In the online version used in this study, the program maintained the structural organization and the theoretical-practical content of the program carried out in person, consisting of eight real-time two-hour group sessions carried out over eight weeks (Table A1). In each session, the instructor guided and carried out formal practices with the participants and facilitated the exchange of experiences. The seven days of the week following each weekly session were dedicated to repeating the formal practices introduced in the respective week with the support of audio guides, the daily record of the experience when training the practices with the support of structured diaries, and implementing informal practices aimed at cultivating the state of mindfulness in day-to-day activities. The program’s practices are characterized by demanding cognitive efforts, especially attention, which progressively evolve over the eight weeks, based on different personal processes focused on: breathing, the body, sensations, sounds, thoughts, and emotions. In order to connect the instructor with the participants in the weekly sessions in real time, a robust video-conferencing platform was used, allowing screen sharing, recording, access via telephone, and uploading to the cloud. Just as it is performed in the face-to-face program, all the support material (audio guides, formal and informal practice diaries, and motivational messages) was sent weekly by e-mail and also through a multi-platform application for instant messaging and voice calls for smartphones, in order to provide the access options, most convenient for each participant.

In summary, the online program differs from the face-to-face one in that the group of participants does not practice in the same physical space and cannot interact face-to-face and share the same environment during the weekly sessions. It still differs by not performing the immersion session in silence for 4 h.

The intervention was applied from April to June 2020 for eight weeks.

### 2.4. Instruments

Information regarding age, sex, racial denomination, profession, marital status, the practice of physical activity, smoking habit, use of alcoholic beverages, number of hours of sleep, and living conditions (alone or accompanied) were collected by a Sociodemographic Data Questionnaire, structured by the researchers. Data related to mental health conditions were collected before and immediately after the end of the intervention using self-applied instruments through electronic forms.

Perceived Stress was assessed by the Perceived Stress Scale—PSS 14, structured by Cohen et al. [21], which was translated and validated for Brazil by Luft et al. [22]. The PSS-14 has 14 questions, with answers ranging from zero to four (zero = never; one = almost never; two = sometimes; three = almost always, and four = always). Scores are obtained by reverse scoring positively stated items (4, 5, 6, 7, 9, 10, and 13) and then summing the scores across all 14 items, including the others with a negative connotation. The total score of the scale can range from zero (no stress) to 56 (extreme stress) [21,22].

Anxiety symptoms were assessed using the Beck Anxiety Inventory (BAI) [23]. This instrument was translated and validated for Brazil by Quintão et al. [24]. It is a self-applied scale of 21 questions, with varied answers: Not at all, Slightly, Moderately, and Severely, to which were assigned scores from 1 to 4, respectively. The final score obtained from the sum of the points can vary between 0 and 63, resulting in the following classification: Up to 10 points indicate an absence of symptoms; 11 to 19 points indicate mild to moderate anxiety; 20 to 30 points indicate moderate anxiety; 31 to 63 points indicate severe anxiety.

Depression symptoms were assessed using the Beck Depression Inventory-II (BDI-II) [25], an instrument translated and validated for Brazil by Gomes-Oliveira et al. [26]. The BDI-II is also self-applied and contains 21 questions, with answers that vary according to the question, from 1 to 3. The final score obtained from the sum of the points can vary between 0 and 63, following the classification: from 0 to 13 points indicates minimal depression; from 14 to 19 points indicate mild depression; from 20 to 28 points indicate moderate depression; from 29 to 63 points indicate severe depression.

Mindfulness was assessed using two instruments, the Mindful Attention Awareness Scale (MAAS) as a unidimensional mindfulness measure and the Five Facet Mindfulness Questionnaire (FFMQ) as a multidimensional mindfulness measure.

The MAAS assesses the attentional aspect of mindfulness, focusing attention on the present moment from a unidimensional perspective. This scale was translated and validated for Brazil by Barros et al. [9] and consists of a self-applied instrument, with 15 items, with response options ranging from 1 = almost always to 6 = almost never, in relation to how much the respondent has experienced what is described in each of the statements. The maximum score that can be reached is 90 points, and the minimum is 15 points, indicating a greater or lesser capacity for mindfulness.

The Five Facet Mindfulness Questionnaire (FFMQ-BR), structured by Baer et al. [27], which was translated and validated for Brazil by Barros et al. [28], measures levels of mindfulness in a multidimensional way, considering five facets. In the Brazilian version, two of the facets of the original version were subdivided, totaling seven, each with different minimum and maximum values, according to the number of questions that characterize it. These are Observing; Describing (subdivided into positive and negative formulation); Acting with awareness (subdivided into autopilot and distraction); Non-reactivity, and Non-judging. The maximum total score is 195 points, and the minimum is 39 points, obtained through the sum of the scores achieved in the facets, indicating the maximum and minimum level of mindfulness, respectively. The analysis of the results of the scale in the present study was performed from the scores of the facets separately, as recommended by the literature [27].

The described instruments had their psychometric properties validated for measuring the respective constructs in the validation studies in the Brazilian population that were referenced.

Participants’ satisfaction with the mindfulness program was assessed using a closed-ended question with five levels for scoring, ranging from “not at all satisfied” to “completely satisfied”. The competencies and skills acquired or improved by the program were accessed through an open question that asked the participant to briefly record their experience. Other assessment items were offered for recording with “yes” and “no” options, relating to improved problem-solving and decision-making skills; practical knowledge of the area of activity, theoretical knowledge of the area of activity; the presence of negative effects resulting from the practice. The feeling of well-being, life satisfaction, and job satisfaction were recorded from three options: greater, equal, and lesser.

### 2.5. Statistical Analysis

Data related to the instruments were analyzed by descriptive and correlational statistics using the IBM^®^ SPSS^®^ Statistics version 25 software. Frequencies, percentages, minimum and maximum values, mean, median and standard deviation were calculated. To compare the variables of interest (perceived stress, depression, anxiety, and mindfulness) measured before and after participation in the online training program for mindfulness practices, the nonparametric Wilcoxon signed-rank test was used to verify differences in the measures repeated in a single sample.

### 2.6. Ethical Aspects

Ethical norms and guidelines were respected, and the project was approved by the Brazilian Ethics Committee (protocol number: 26631119.5.0000.5393) in accordance with the Code of Ethics of the World Medical Association (Declaration of Helsinki) and the Resolution 466/12 of Brazilian legislation.

## 3. Results

### 3.1. Adherence to Treatment

Adherence to treatment, defined as the percentage of participants who completed the mindfulness program, was 70.12% (54/77). Among the participants who were informed of the reasons for discontinuing participation (21/23), 91.47% reported that it was not possible to conciliate participation in the program with work activities. Participants who did not complete the post-intervention measurement were 18.51% post-test (10/54). Thus, 44 nurses concluded the intervention program and the necessary assessments to complete the study.

### 3.2. Sample Characterization

The mean age of the sample was 33.8 years (SD = 5.6), and women represented 84.1% of the participants. Regarding education, 77.3% had a postgraduate degree, 20.4% were undergraduates, and 2.3% had completed high school. Having children was reported by 61.4%, 90.9% reported having a religious belief, and 54.5% reported practicing a religion. The participants worked an average of 47.2 h per week; 72.7% had an employment contract, 20.5% had two different jobs, and 6.8% had three or more (Table 1). The professionals participating in the study reported sleeping an average of 6.4 h per day and classified their sleep as very good (4.5%), good (40.9%), satisfactory (34.1%), or poor or very poor (18.2%), 75.0% practiced physical activity; 95.5% did not use tobacco; 54.5% consumed alcohol on average 1.6 times a week. Regarding medical aspects, 54.5% underwent medical consultation; 13.6% used psychotropic drugs; 38.6% needed psychological care in the last year; and 20.5% had some chronic disease. Concerning the COVID-19 pandemic, 95.5% of the professionals participating in the study indicated that their work routines were impacted, 88.6% felt emotionally impacted, and among them, 80.9% considered that this emotional impact could affect their clinical practice.

### 3.3. Intervention Effectiveness

Table 2 presents the mean and standard deviations of the outcome measures of this study. The perceived stress of nurses was significantly reduced after participating in the mindfulness program. Improvements in depression and anxiety scores were also observed, with significantly lower means measured after the intervention. The unidimensional mindfulness measure increased significantly, revealing the primary effect of the practices implemented in the intervention.

The results of the multidimensional mindfulness assessment (FFMQ) showed a significant increase in six of the mindfulness components that correspond to the facets of non-judging of inner experience; acting with awareness–autopilot; observing, describing–positive formulation; non-reactivity to inner experience; and acting with awareness–distraction. Only one of the facets, “describing–negative formulation”, showed no difference between the assessments before and after participating in the mindfulness-based intervention (Table 3).

### 3.4. Participants’ Satisfaction with the Intervention

All participants (100.0%) scored the maximum level of satisfaction with the program and would recommend it to other professionals. Regarding the main competencies and skills acquired or improved by the program, 93.18% reported relief from the physical and mental overload in the face of the challenges and changes imposed by the COVID-19 pandemic, while 88.67% had improvement in problem-solving and making a decision, 79.54% in practical knowledge for the area of activity and 40.90% in theoretical knowledge for the area of activity (40.90%). All participants (100.0%) reported that they did not experience negative effects resulting from the practices. They indicated a greater sense of well-being (97.72%), greater satisfaction with life (90.90%), and greater satisfaction with work (86.36%) when completing the program. None of the participants indicated lower values of well-being, life satisfaction, and job satisfaction.

## 4. Discussion

The rapid spread of COVID-19 has placed an enormous burden on healthcare systems around the world. The effects on frontline nursing professionals were also severe. Nurses are one of the groups most at risk of infection due to prolonged contact with clients, and the negative psychological effects of working on the front lines of the pandemic were also significant [29]. In this sense, the results of a systematic review of the literature, including 91 studies, showed that in nurses, the prevalence of depression and anxiety was 35% and 37%, and of stress and sleep disorders was 43% and 43% [30], respectively. These findings highlight the importance of providing comprehensive support strategies to reduce the psychological impact of the COVID-19 pandemic among nurses. We recognize the challenge faced by nursing and medical teams and the need for self-care and reducing the risk of mental illness inherent in this context.

The results of the present study highlighted the viability and acceptability of the mindfulness program for the investigated nursing professionals. Adherence was 70% of the participants, a result similar to that found in a study that evaluated Mindful Self-care and Resiliency (MSCR), an intervention applied to Australian nurses [31]. The difficulty in reconciling participation in the program with work activities, which was reported as the main reason for participants who did not conclude the intervention, reveals the need to critically assess the support needed within the leadership to guarantee the necessary conditions for effective participation of the participants. A similar exploratory study with frontline health professionals carried out in Spain found that establishing the goal of self-care and offering it to all professionals in the workplace can be factors that facilitate participation and contribute to reducing the stigmatization of workers most in need of mental health support [32].

Nursing professionals reported a higher level of satisfaction with life and work when participating in the program and reported that they did not experience negative effects resulting from the practices, similar to what was reported in another study with health professionals [33]. Similarly, a study conducted by Fortney et al. [34] demonstrated that participants described high levels of satisfaction with life and work after completing a mindfulness program. The results of the assessment of satisfaction and the occurrence of negative experiences during the program contribute to indicating the intervention as a useful and safe measure. In this sense, we highlight the particularity of the context of the study developed in the midst of considerably adverse circumstances imposed by the pandemic (constant changes, shortage of materials and equipment, overload due to long working hours, risk of infection, etc.), which adds greater relevance to the results achieved.

Levels of stress, anxiety, and depression were also evaluated in this study, and the results point to a significant reduction in the levels of the three variables measured. In general, the results found in this study are similar to the findings of another study carried out in a Brazilian hospital that aimed to evaluate the effects of a face-to-face Stress Reduction Program, including mindfulness and meditation, among nursing professionals working in a hospital environment. Analyses revealed a significant reduction between pre-intervention and post-intervention scores for stress, burnout, depression, and anxiety. These variables did not show significant differences between the post-intervention and follow-up scores, which indicates that there was a maintenance of the results after six months of the intervention [35].

With regard to stress, several studies indicate the effectiveness of mindfulness practices in improving stress [6,31,34]. In an extensive literature review, Lomas et al. [6] found that most of the 41 included studies showed a statistically significant reduction in stress levels. Similar results were also found in a study evaluating mindfulness practices with nursing professionals [31] and primary care professionals [34]. In the present study, the reduction of stress after carrying out the intervention in the online format and in the pandemic context expands the possibilities and relevance for the implementation of this type of therapeutic resource.

Mindfulness practices help reduce stress by altering brain structures and activity in regions associated with attention and emotional regulation [36,37]. In a review of mindfulness studies [38], strong evidence was found that people who participated in mindfulness practices were less likely to react with negative thoughts or reactions and negative emotions in times of stress. There was also moderate evidence that people who participated in the interventions were better able to focus on the present and less likely to worry and dwell on negative thoughts or experiences.

With regard to depression and anxiety, the results of this study demonstrated a significant reduction in symptoms after participating in mindfulness practices. Studies suggest that mindfulness-based cognitive therapy is as effective as medication in preventing depression relapse among adults with a history of recurrent depression and in reducing depressive symptoms among those with active depression [39,40]. Nursing professionals have a high prevalence of depression and anxiety, and it is important to consider their mental health for the sustainability of care capabilities. A literature review conducted by Van der Riet and colleagues [41] evaluated 16 international studies and found a significant reduction in levels of depression and anxiety in nursing professionals after participating in mindfulness programs. These results are in line with the findings of this study, reaffirming the importance of mindfulness practices for nursing professionals in the treatment and prevention of depression and anxiety.

The Mindful Attention Awareness Scale (MAAS) assesses a core feature of dispositional mindfulness, namely, open or receptive awareness and attention to what is happening in the present. Its application in the present study revealed that the participants showed a significant increase in the scale scores, which indicates an increase in the ability to concentrate on conscious awareness and living in the present moment. Similarly, a study conducted with Chinese nursing professionals participating in a mindfulness practice program found a significant improvement in mindfulness levels after completing the program when compared to the control group [42]. In the same vein, Horner et al. [43] carried out a study that aimed to explore the impact of mindfulness training for nursing staff on levels of mindfulness, compassion, burnout, and stress. Results showed that the intervention group showed increased levels of mindfulness and compassion while decreasing levels of burnout and stress after training when compared to the control group.

Mindfulness can also be conceptualized as a multifaceted construct consisting of several related skills [28]. Observing is the tendency to perceive or attend to internal and external experiences. Describing involves labeling observed experiences with words. Acting mindfully refers to paying attention to the activity at hand and is often contrasted with behaving mechanically while attention is focused elsewhere (often called autopilot). Not judging inner experience involves taking a non-evaluative stance toward thoughts and emotions. Non-reactivity to inner experience is the tendency to allow feelings and thoughts to come and go without getting carried away or becoming caught up in them [44].

The results of the present study demonstrated a significant increase in levels of mindfulness in six of the seven facets studied. Only the facet, “Describing—negative formulation”, did not show a statistically significant change. This result may be related to the pandemic context that may have made it difficult to increase this component of mindfulness. Scientific evidence revealed that negative emotions were altered in the COVID-19 pandemic; the prevalence of sadness, worry, stress, and anger fluctuated considerably over time and steadily declined to pre-pandemic levels in mid-2021 [45].

Similar results were found by Yang et al. [46] in a randomized clinical trial performed with medical students. The Describing and Non-judging of inner experience facets did not show significant differences before and after the intervention. Yamada and colleagues [47] implemented a mindfulness program in which participants in the mindfulness group, on average, increased their scores on mindfulness facets by nearly 9% compared to the control group. Greeson and colleagues [48] implemented a mindfulness intervention in college students and found significant increases in mindfulness and large effect sizes at the end of the intervention compared to a waitlist control group.

The study’s novelty was to highlight the usefulness of the eight-week mindfulness-based online training as a support for nursing professionals in the context of the mental health risk of the COVID-19 pandemic. It is limited by the absence of a comparison group and a prospective way of evaluating mental health variables. The follow-up of the participants could bring evidence about the maintenance of changes in the mental health status observed after the intervention. Additionally, the before-after design used in the present study limits the observation that factors other than the intervention contributed to changes in outcome measures. However, the findings are encouraging and indicate that the online training program for mindfulness practices can help nurses maintain mental health in challenging contexts, such as the COVID-19 pandemic, as it reduces perceived stress and symptoms of depression and anxiety, as well as increases the level of mindfulness. The results achieved with the online program were the same as those found in the face-to-face program, corroborating the recommendation to implement the practice also in the online format.

We emphasize that the sample analyzed in the present study potentially had some level of motivation to participate in the intervention, whether due to the need for psycho-emotional support or personal interest in the practice. Thus, we recognize the caution regarding the generalization of our results and recommend the development of studies on the effect of motivating factors in health professionals on the results obtained with training in mindfulness practices.

## 5. Conclusions

Faced with the changes resulting from the pandemic, the imposed need to offer support to mental health in an online format allowed us to evaluate the usefulness of an online program of mindfulness practices offered to help health professionals deal with stress in the challenging context of the COVID-19 pandemic. This resulted in the identification of a decrease in perceived stress, symptoms of anxiety and depression, an increase in the levels of uni- and multidimensional mindfulness, and greater satisfaction of the participants with life and work. In short, our results indicate that mindfulness-based interventions represent a useful strategy for nursing professionals in the face of the need for self-care with mental health and mechanisms that guarantee the sustainability of their abilities to continue providing health care.

## Figures and Tables

**Figure 1 ijerph-20-03666-f001:**
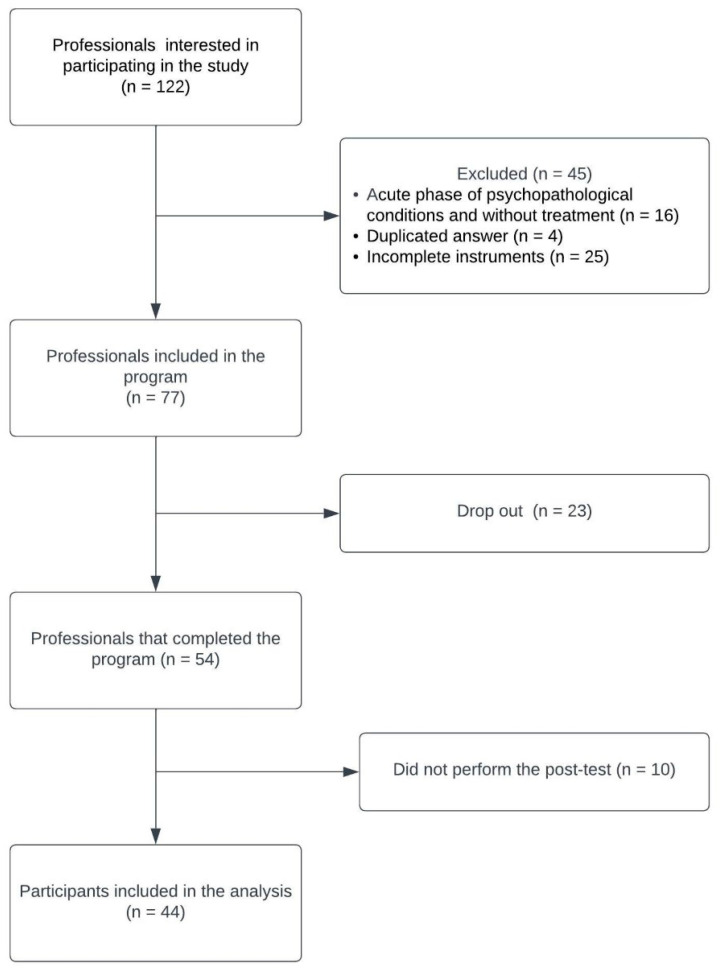
Flowchart of the recruitment of nursing professionals and data collection.

**Table 1 ijerph-20-03666-t001:** Sociodemographic characteristics and working and health conditions of nursing professionals participants of the study (n = 44).

Variable	Value
Age, years [mean (SD ^1^)]	33.8 (5.6)
Gender [n (%)]	
Female	37 (84.1)
Male	7 (15.9)
Has children [n (%)]	27 (61.4)
Educational level [n (%)]	
High school	1 (2.3)
Undergraduate	9 (20.4)
Post-graduated	34 (77.3)
Weekly work, hours [mean (SD^1^)]	47.2 (13.1)
Number of jobs [n (%)]	
One	32 (72.7)
Two	9 (20.5)
Three or more	3 (6.8)
Has some religion belief [n (%)]	40 (90.9)
Practices religion [n (%)]	24 (54.5)
Physical exercise [n (%)]	33 (75.5)
Tobacco use [n (%)]	2 (4.5)
Alcohol use [n (%)]	24 (54.5)

^1^ SD: standard deviation.

**Table 2 ijerph-20-03666-t002:** Comparison of perceived stress, anxiety, depression, and unidimensional mindfulness presented pre- and post-mindfulness-based intervention by nursing professionals (n = 44).

Variable	Pre intervention Mean (SD ^1^)	Post Intervention Mean (SD ^1^)	Wilcoxon Signed Rank Test (*p*)
Perceived Stress (PSS ^2^)	29.5 (7.0)	22.5 (5.7)	Z = −4.25; *p* < 0.001
Anxiety (BAI ^3^)	35.0 (9.2)	28.0 (8.5)	Z = −3.16; *p* = 0.002
Depression (BDI ^4^)	11.0 (7.1)	6.5 (5.4)	Z = −3.50; *p* < 0.001
Mindfulness (MAAS ^5^)	47.0 (13.6)	55.0 (12.3)	Z = 3.92; *p* < 0.001

Alpha criteria = 0.05; ^1^ SD: standard deviation; ^2^ PSS: Perceived Stress Scale; ^3^ BAI: Beck Anxiety Inventory; ^4^ BDI: Beck Depression Inventory; ^5^ MAAS: Mindful Attention Awareness Scale.

**Table 3 ijerph-20-03666-t003:** Comparison of the levels of multidimensional mindfulness presented pre- and post the mindfulness-based intervention by nursing professionals (n = 44).

Variable	Pre Intervention Mean (SD ^1^)	Post Intervention Mean (SD ^1^)	Wilcoxon Signed Rank Test (*p*)
Facet 1—Non-judging of inner experience	26.0 (6.6)	29.0 (7.6)	Z = 3.18; *p* = 0.001
Facet 2—Acting with awareness–autopilot	16.5 (3.2)	18.5 (2.7)	Z = 3.33; *p* = 0.001
Facet 3—Observing	23.5 (7.5)	29.5 (6.1)	Z = 4.81; *p* < 0.001
Facet 4—Describing–positive formulation	13.0 (5.5)	17.0 (4.4)	Z = 3.97; *p* < 0.001
Facet 5—Describing–negative formulation	12.0 (3.4)	12.0 (2.8)	Z = 1.88; *p* = 0.059
Facet 6—Non-reactivity to inner experience	16.0 (6.1)	21.0 (5.4)	Z = 2.80; *p* = 0.005
Facet 7—Acting with awareness–distraction	10.5 (3.4)	12.0 (2.7)	Z = 3.47; *p* = 0.001
Total Score	111.0 (23.2)	137.5 (20.7)	Z = 4.85; *p* < 0.001

Alpha criteria = 0.05; ^1^ SD: standard deviation.

## Data Availability

The data presented in this study are available on request from the corresponding authors. The data are not publicly available due to ethical reasons.

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
