# Peer review of "The Impact of an Online Mindfulness-Based Practice Program on the Mental Health of Brazilian Nurses during the COVID-19 Pandemic"

_ijerph, 2023, doi:10.3390/ijerph20043666_

Round 1

Reviewer 1 Report

Dear authors,

1.General comments

This intervention study evaluated the usefulness of an online mindfulness practice program designed to help healthcare professionals in Brazil cope with stress during the Covid-19 pandemic. The results showed that the program was effective, with reductions in perceived stress, anxiety, and depression. The mental burden on healthcare professionals who deal with Covid-19 patients is significant, and the usefulness of an easily accessible online version of the mindfulness practice program is an interesting public health finding. In addition, the results of this study may lead to the expansion of  an online mindfulness practice program . However, the study has several areas for improvement at this time.

2.Specific comments

a)Major

ⅰ)

The authors used a program with a well-established methodology and stated that the online practice program intervention was as effective as the face-to-face version. However, due to the study design, which has no comparators, you have not adequately validated the effectiveness. Therefore, the word "effect" is an oversimplification and should be changed to "usefulness. You should also illustrate the existing program content and explain the differences between the face-to-face and online versions. This will make it easier for readers to understand the program content. In addition, the need for a practical online version of the program should be added in the introduction. Also, is this program specific to Covid-19? You should provide a clear explanation on this point as well.

ⅱ)

You should list the type of occupation of the study participants in Table 1. The introduction and methods mention physicians and other health care professionals, but the discussion focuses only on nurses. Therefore, it is not an adequate discussion. I suggest that the results and discussion would be clearer if they were reanalyzed using only nurses. If you do not focus on nurses, add citations and discussion of studies conducted on medical doctors and other health care professionals.

bMinor

ⅰ)

It would be easier to understand if you would attach a flowchart of the study participants.

ⅱ)

The authors should describe the recruitment method and sample in more detail. For example, what medical institutions and how many locations did they mail to? Also, you should describe what type of social networking sites you used and how long you recruited for participation.

ⅲ)

The authors should be more specific about one nurse who intervened with the participant, including her nursing history.

ⅳ)

I suggest that the p-value in the Table be p<0.001 instead of p=0.000. Also, the numbers should be standardized to one decimal place.

ⅴ)

Since this is a result of repetition, delete ”The indication of not 13.64%, respectively.” in the last paragraph of 3,3

ⅵ)

The authors need to specifically describe the limitations of how the prospective method of assessing mental health variables impacts the results.

Author Response

Dear Reviewer

We appreciate your careful review of our manuscript. Please see the attached file for our comments.

Best regards,

Reviewer 2 Report

The authors conducted a before-after study to examine the feasibility and effectiveness of an online mindfulness practices program on the mental health of nurses and physicians during the Covid-19 pandemic. By analyzing the data of 44 participants, the authors showed that perceived stress, depression, anxiety, and mindfulness significantly improved after the intervention. In addition, high satisfaction was observed. Given the growing mental health problems of healthcare professionals during the pandemic, the findings of this study are significant and could have implications for future research directions.

Comments:

1.      Materials and Methods (2.2. Recruitment and Sample): “At the end of the 8-week intervention, 54 professionals completed the program, and 44 professionals performed the post-test, --.” In other words, 33 enrolled participants were lost to follow-up and failed to contribute to the data analyzed. What were the baseline characteristics of these 33 participants? Were there differences between these 33 participants and the 44 participants who completed the post-test? A presentation of the data and the results of statistical tests is recommended.  

2.      Materials and Methods (2.4. Instruments): “Data related to mental health conditions were collected before and immediately after the end of the intervention using five instruments.” Was the assessment of the mental health outcomes conducted by the research personnel or via a self-administered questionnaire (in electronic forms)? If the research personnel conducted it, were the research personnel who assessed the outcome measures blinded? A more detailed description is necessary.

3.      Materials and Methods (2.5. Statistical Analysis): “-, the non-parametric Wilcoxon rank sum test was used to verify differences in the -.” However, Tables 2 and 3 showed the Wilcoxon signed rank test results. Was the “Wilcoxon signed rank test” or “Wilcoxon rank sum test” used? Please clarify.

4.      Materials and Methods (2.5. Statistical Analysis): Please describe the methods, if any, has been used to determine the sample size.

5.      Results (3.3. Intervention effectiveness): “Table 2 presents the mean and standard deviations of the outcome measures of this study.” However, Table 2 shows the “median.” Was it “mean” or “median”? Please clarify. In addition, if it were “median,” a presentation of the range, instead of the standard deviation, would be more appropriate.

6.      Discussion: A major limitation of this study lies in the before-after design. Using such a design, the possibility that factors other than the intervention have caused the changes in outcome measures could not be excluded. A discussion of this limitation is recommended.

7.      Discussion: As reported in the Materials and Methods (2.2. Recruitment and Sample), “this is a non-probabilistic convenience sample, which included: nursing professionals (nurses, technicians and assistants) and physicians, who assisted in any health services in Brazil during the Covid-19 pandemic and expressed interest to participate in the online training program--.” Therefore, this study sample quite likely consisted of healthcare professionals who showed great motivation, and they may be different from the target population. Would this affect the generalizability of the results? A discussion of the potential implications regarding the generalizability of the results is encouraged.  

Author Response

Dear Reviewer

We appreciate your fruitful suggestions for the improvement of our manuscript. Please see the attached file for our comments.

Best regards,

Reviewer 3 Report

Thanks for the opportunity to review this interesting paper. This is a quantitative, before-after study, developed to evaluate the effects of an online mindfulness practices program designed to help healthcare professionals deal with stress in the challenging context of the Covid-19 pandemic, through the assessment of perceived stress, anxiety and depression, levels of mindfulness; and participants' satisfaction with the program. The results indicated that results indicate that mindfulness-based interventions represent an effective strategy for health professionals in the face of the need for self-care with mental health and mechanisms that guarantee the sustainability of their capacities to continue exercising health care. Overall, this manuscript is well written. I only have a few comments.

1. In the introduction, please describe the results of mental health investigation of health workers worldwide first, and provide a review of existing psychosocial intervention to reduce psychological burden of the health workers.

2. In 2.2, please give rationale for paricipants exclusion.

3. Please give psychomatric properties of the measurement tools.

4. In the discussion, please make it concise. highlight the results, exaplian the results and provide implications for future practice and research.

Author Response

(The authors gave the same response as above.)

Reviewer 4 Report

This manuscript assessed whether an on-line mindfullness program is capable of improving the mental health in nurses and physicians. The study is interesting but could be improved:

- Most of subjects were female. I would recommend to emphasize it on the title or even remove the male subjects to avoid biased analysis.

- Authors considered only subjects free of acute psychopathological conditions. However, they did not assess subjects previous psychopathological status. This this could generate biased conclusions. Please, address this issue.

- Change Table 2 and 3 into graphs would be great. It would become data easier for the reader.

- Authors described several aspects of minfullness in the Discussion section. Please, place it on the materials and methods section.

- It is not clear for this reviewers how the program was carried out. Please, make a graph summarizing the methodology adopted. What was the content used in the online activities ? Please, provide more details.

-Authors must state in the title that only brazilian subjects were considered in this study.

- I would be interesting to discuss whether these conclusions would differ from a non-pandemic scenario. 

- Authors compare their findings with other studies which assessed the mindfullness. However is not clear their novelty. Please, address.

- Please, insert in the discussion section whether the use of alcohol and tobacco as well as exercise or religion could impact the findings. This is very important since these factors would sharply interfere in anxiety, depression and stress.

Author Response

(The authors gave the same response as above.)

Round 2

Reviewer 1 Report

Dear authors

I have reviewed the revised version. Thank you for accepting my suggestion. I think it is now much easier to understand.

Best regards.